# Environmental Factors Determining Body Mass Index (BMI) within 9 Months of Therapy Post Bariatric Surgery—Sleeve Gastrectomy (SG)

**DOI:** 10.3390/nu14245401

**Published:** 2022-12-19

**Authors:** Agata Wawrzyniak, Monika Krotki

**Affiliations:** Department of Human Nutrition, Institute of Human Nutrition Sciences, Warsaw University of Life Sciences (WULS-SGGW), 02-776 Warsaw, Poland

**Keywords:** SG bariatric surgery, dietary care, adults, anthropometric measurements, blood pressure, lifestyle, eating habits, BMI variation

## Abstract

Treatment of obesity should be multidirectional and include, in addition to bariatric surgery, changing the key factors of lifestyle and eating habits. The study aimed to assess the impact of bariatric surgery and dietary care on anthropometric measurements, blood pressure, changes in lifestyle, and eating habits of patients within 9 months after the procedure, with the selection of environmental factors determining BMI variation. The study included 30 SG patients before surgery (month zero) and at 1, 3, 6, and 9 months after SG. Patients completed a questionnaire regarding age, sex, place of residence, education, professional activity, number of family members, financial situation, family history of obesity, previous forms of therapy, self-assessment of nutritional knowledge, receiving and following nutritional recommendations, eating habits, frequency of body weight control, leisure time. Body weight, height, waist and hip circumference, and systolic and diastolic pressure were measured, and BMI and WHR (Waist to Hip Ratio) were calculated. Within 9 months after the procedure, the patients’ body weight and BMI decreased on average by 26%. Post bariatric surgery, patients changed their eating habits. The influence of bariatric SG surgery and time after surgery was decisive for the normalization of BMI and explained the 33% variation in BMI up to 9 months after the procedure. Other factors important for the normalization of BMI after surgery were: male gender, older age of patients, family obesity (non-modifiable factors), as well as previous forms of therapy related to weight loss before surgery, shortening the intervals between meals and stopping eating at night (modifiable factors). The tested model explained 68% of the BMI variation after SG surgery for all assessed factors. Changes in lifestyle and eating habits in bariatric patients are crucial to maintaining the effect of bariatric surgery.

## 1. Introduction

Obesity is a chronic metabolic disease of complex etiology that requires treatment, including surgery. Obesity treatment is multidirectional and requires the cooperation of many specialists (internist, surgeon, physiotherapist, psychologist, dietitian). It consists in introducing changes in the external environment (lifestyle, spending leisure time), behavioral changes (psychotherapy, nutritional education), dietary treatment (introducing controlled consumption of food and energy intake), introducing controlled physical activity [1,2].

The development of civilization meant that the current lifestyle requires less energy expenditure. All communication facilities, mechanization in the household, and other organization of work and leisure limit physical activity, which may delay body weight loss after bariatric surgery and its effectiveness. New forms and types of food are also conducive to increasing consumption and perpetuating an unfavorable energy balance. However, other factors are less studied in the combined effect on anthropometric measurements, including BMI, after SG bariatric surgery, i.e., socio-demographic factors (age, sex, place of residence, level of education, professional activity, financial situation), family conditions (number of family members, family history of obesity), the impact of nutritional knowledge and eating habits (number of meals, breaks between meals, control of food consumption, snacking during the day, eating at night, alcohol consumption), forms of therapy (trials of therapy before bariatric surgery, body weight control before and after surgery). Some of the above factors can be modified by patients in the treatment process; however, there is a lack of studies providing knowledge in this field, including the impact of the patient’s environment, with its cumulative assessment, on body weight loss after SG bariatric surgery and the improvement of body mass index compared to reference values. Previous studies have examined several factors that influence body weight or BMI after bariatric surgery [3,4,5,6,7,8]. Hence the need to assess which of the selected environmental factors influence, in the cumulative assessment, the success of body weight loss after SG bariatric surgery and the normalization of BMI in patients undergoing dietary care (necessary in the therapeutic process) while assessing their susceptibility to modification.

The authors put forward a research hypothesis that the lifestyle of patients undergoing bariatric surgery and postoperative dietary care changes, and environmental factors, after the surgery has a significant impact on anthropometric measurements, including BMI variation within 9 months after surgery.

## 2. Materials and Methods

### 2.1. General Information

The study followed patients of the General, Oncological and Digestive Tract Surgery Department at the Medical Centre of Postgraduate Education at Orlowski Hospital in Warsaw, Poland. The study received the consent of the Bioethical Commission of the Medical Centre for Postgraduate Education (Warsaw, Poland) on 12 April 2017 (KB-W-382/2017) and the individual consent of patients [9,10].

### 2.2. Study Participants

The study included patients treated surgically for obesity class III and II with comorbidities, such as heart disease, metabolic disorders, lipid disorders, diabetes, sleep apnea and osteoarthritis. The patients met the following detailed criteria qualifying for the operation: an age range between 18 and 60 years, BMI ≥ 40 kg/m^2^ or BMI 35–39.9 kg/m^2^ in persons additionally experiencing comorbidities. The exclusion criteria were as follows: inflammatory bowel disease, chronic oesophagitis, gastric and duodenal ulcers constituting a risk of gastrointestinal bleeding, as well as digestive tract anomalies, severe heart disease and breathing difficulties, alcohol abuse, drug addiction, pregnancy, mental disorders, personality disorders, severe depression, a possible lack of patient engagement in the post-surgical treatment process, an inability to look after oneself and a lack of due medical care from a caregiver post-surgery [11,12,13]. The study followed SG patients who regularly participated in the study, i.e., before the procedure (month zero) and in the 1st, 3rd, 6th and 9th month after bariatric surgery (irregular patients, although qualified, were not included).

Throughout the postoperative follow-up period (including visits at 1, 3, 6, and 9 months after SG surgery), all patients received dietary recommendations from a certified dietitian at a dietary clinic in the form of oral recommendations, a brochure or leaflet, and a ready-made menu (recommended and non-recommended products, quantitative nutritional guidelines, information on culinary techniques).

### 2.3. Socio-Demographic Factors

Participants in each study period completed a questionnaire on age, sex, place of residence, education level, professional activity, number of family members, financial situation, and family history of obesity (parents, grandparents), selecting the answer from the proposed cafeteria. Before the procedure, the respondents answered questions on whether the surgery was the first or subsequent treatment attempt and what previous forms of therapy they had undergone.

### 2.4. Self-Assessment of Nutritional Knowledge and Eating Habits

At each stage of the study, patients were asked to self-assess their nutritional knowledge (bad, neither good nor bad, good) and their willingness to expand it. Before the procedure, patients were asked about the sources (doctor, nurse, dietitian), form (oral, brochure/leaflet, menu), and the content of the nutritional recommendations received (recommended and non-recommended products, quantitative nutritional guidelines, information on culinary techniques, ready-made menus) and the degree of adherence to dietary recommendations. Questions about the degree of compliance with dietary recommendations were also repeated after surgery. At each stage of the study, patients were asked about the number of meals consumed, breaks between meals in hours, control of food consumption according to the recommendations, snacking, eating at night, the most commonly used culinary techniques and alcohol intake. Questions on eating habits were developed on the basis of recommendations provided to patients in the form of educational materials or orally. In addition, the patients’ dietary intake was assessed with a 4-day food record covering three working days and one non-working day [9,10]. Patients reported all consumed products, meals and drinks in each study period. At each control visit, the patients provided the trade name of the dietary supplement they were taking and the dose. Based on a comprehensive assessment [9,10], the patients received nutritional recommendations for the next period from a dietitian.

### 2.5. Body Weight Control, Leisure Activity

Patients declared whether and how often they controlled their body weight and what motivated them to do so. Patients also confirmed how they spend their leisure time (passively/actively, e.g., aerobics, walking, swimming, cycling, gymnastics).

### 2.6. Anthropometric Measurements and Measurements of Systolic and Diastolic Blood Pressure

Body weight was measured according to the procedure using a MENSOR WE150P3M (X) electronic scale (Mensor Corporation, Warsaw, Poland), and the height of patients was measured with a SECA 213 device (Seca, Hamburg, Germany) [14]. Based on the height and weight measurements, the body mass index (BMI) in kg/m^2^ was calculated and compared with the reference values [8]. Waist and hip circumferences were measured according to the procedure with a medical tape measure. Waist circumferences over 80 cm for women and 94 cm for men were classified as abdominal obesity. The results of the measurements were used to calculate the WHR index as the waist (cm)/hip (cm) quotient. The WHR reference value for women was set at ≤0.80 and for men at ≤1.00 [15].

Blood pressure was measured at each morning visit in a sitting position, according to the procedure provided by the manufacturer. An OMRON M3 upper arm blood pressure monitor (Omron Corporation, Kyoto, Japan) was used for the measurement. Measurements were made on the left arm each time. Optimal systolic blood pressure < 120 mmHg and diastolic blood pressure < 80 mmHg were used as reference values [16,17].

### 2.7. Statistical Analysis

Statistical analysis was performed using the IBM SPSS Statistics25 software package (IBM Corp., Armonk, NY, USA). Descriptive statistics and testing of the normality of the distribution of continuous variables were performed using the Shapiro-Wilk test. The results are presented as mean values, standard deviations, and percentages, depending on the type of the variable. The non-parametric Friedman test was used to verify the significance between groups in repeated measurements, and the W-Kendal test was used for the nominal variable for repeated measurements. Linear regression models were tested to evaluate the effect of selected nutritional and non-nutritional factors, as well as modifiable and non-modifiable, on BMI variation in bariatric patients within 9 months of SG surgery. The value of α = 0.05 was considered statistically significant.

## 3. Results

### 3.1. Characteristics of the Studied Group

The study involved 30 patients (24 women and 6 men) who underwent SG bariatric surgery. The characteristics of age, sex, place of residence, level of education, professional activity, financial situation, and the number of family members are presented in Table 1. The average age of the patients was 45 years, the youngest patient was 18 years old, and the oldest patient was 60 years old. Most of the respondents lived in the city/town, in 2- or 4-person families, had a secondary education, and worked professionally, with good or neither good/nor bad financial situations. Patients declared obesity in their family (83% of parents with obesity and/or grandparents with obesity); for 93% of patients, it was another attempt at therapy after previous diet or diet and drug therapy.

### 3.2. Dietary Recommendations, Self-Assessment of Nutritional Knowledge

Before the procedure, 80% of respondents received dietary recommendations, mainly from a dietitian (70%) and mostly in the form of a brochure/leaflet (70%) or orally (53%) (Table 2). After the SG surgery, all patients who came to the dietary clinic received a full package of oral recommendations, brochures/leaflets, a ready-made menu regarding recommended or non-recommended products, quantitative nutritional guidelines, and culinary techniques. These changes were quantitatively and qualitatively significant for patients before and after bariatric surgery. Moreover, in the period before bariatric surgery and the 9th month after the procedure, approximately 40% of patients did not adjust or slightly adjusted to the recommendations; in the first month after the surgery, 84% of respondents adjusted to a great extent or completely. Throughout the study, patients were willing to broaden their nutritional knowledge (87–97%), and their nutritional knowledge increased significantly during therapy.

### 3.3. Patients’ Eating Habits, Body Weight Control, Leisure Activities

After bariatric surgery, patients changed their eating habits, dividing their food into more meals and eating more often according to dietary recommendations. They controlled the consumption of food more often, snacked and fried less often, and boiled/steamed more often. They slightly reduced eating at night (from 17 to 3% of respondents). In the postoperative period, they drank alcohol less frequently. They controlled their body weight more often, but the reasons for weight control did not change, and they spent their leisure time as they did before the surgery.

### 3.4. Changes in Anthropometric Measurements and Blood Pressure in Patients within 9 Months of Therapy

Within 9 months after the procedure, patients’ body weight and BMI decreased by an average of 26%, waist circumference by 21%, hip circumference by 19%, WHR decreased by 4% and diastolic blood pressure by 6% (Table 3). At the end of 9 months, 10% of the respondents had a normal BMI, and as many as 30% were overweight instead of obese (Table 4). The percentage of patients with class III obesity (BMI ≥ 40) decreased from 66% to 7%. The WHR index normalized in 27% of the respondents, and the systolic and diastolic blood pressure normalized in 20%.

### 3.5. Environmental Factors Determining BMI in Patients within 9 Months of Therapy

In model 1, examining only the effect of SG surgery within 9 months on changes in patients’ BMI, it was estimated that the studied variable accounted for 33% of the variation in postoperative body mass index (R^2^ = 0.334) (Table 5).

In model 2, taking into account all the examined factors, it was estimated that the changes in BMI over the 9-month period were significantly influenced by: time after surgery, sex, age, obesity in the family, the number of previous therapies, breaks between meals, eating at night, and the study model explained almost 68% of the BMI variation (R^2^ = 0.676). The decrease in BMI up to 9 months after surgery was systematic; older patients easily normalized their BMI, including men and patients undergoing preoperative therapy (diet/medications). Patients with a family history of obesity were more determined to normalize their BMI during treatment. Moreover, the reduction of BMI was positively influenced by shorter intervals between meals and a reduction in night eating.

## 4. Discussion

In the conducted research, scientists confirmed that patients who underwent bariatric surgery and dietary care after the surgery changed their lifestyle and eating habits. In accordance with dietary recommendations, patients ate meals more often, with shorter breaks between meals, and controlled their food consumption, including snacking during the day. Patients changed their cooking techniques, limited those with too much fat (frying), reduced their alcohol intake, and controlled their body weight more frequently.

In addition, the researchers confirmed that the influence of SG bariatric surgery and the time following surgery was decisive for BMI normalization and explained 1/3 of the variation in BMI up to 9 months after surgery. Other factors important for BMI normalization after surgery were male gender, older age of patients, family obesity (non-modifiable factors), and the number of weight loss procedures before surgery (modifiable factor). The study also found that shortening the interval between meals and stopping eating at night positively affected BMI changes in the study group, accounting for nearly 68% of the variation in BMI after SG surgery for all factors assessed.

### 4.1. Assessment of Changes in Anthropometric Measurements and Blood Pressure after Bariatric Surgery

Many studies have documented the undisputed benefits of SG, compared to conservative treatment, for body weight loss in patients with obesity [18,19,20,21]. In our study, patients repeatedly attempted to lose body weight, in most cases using a diet. These attempts were unsuccessful, and the patients underwent bariatric surgery. Most studies observed the greatest weight loss in the first and second year after surgery, especially in the first 3–6 months [18,22,23,24,25,26]. Long-term observations are less numerous [23,27,28,29]. Most studies have shown a trend for postoperative body weight gain in patients 2 years after surgery as a result of improved food tolerance and increased stomach volume. Additionally, over time, fewer patients took part in the long-term follow-up, which could have influenced the results of the studies [22,23,30,31,32].

In our study, changes in body weight classification according to BMI were observed. Similar results were obtained by Mohammed et al. [33], who observed a statistically significant reduction in BMI 12 months after SG compared to the period before the procedure. Gjessing et al. [34] documented a decrease in waist circumference by 14% after 3 months and by 24% after 12 months after SG; however, none of the subjects reached the reference values. Albanopoulus et al. [22] proved in SG patients a reduction in the percentage of respondents requiring treatment for arterial hypertension from 33% to 15% 1 year after surgery. Major et al. [35] found that after 1 year after SG, nearly 30% of patients did not require treatment for arterial hypertension. Sarkhosh et al. [36] observed the resolution of arterial hypertension in 58% of patients after SG. On average, 75% of patients experienced normalization or partial reduction in blood pressure. The differences in the obtained results stemmed from the fact that the study by Sarkhosh et al. [36] adopted 140 mmHg as the reference value for systolic pressure and 90 mmHg for diastolic pressure. In our study, the reference value of systolic blood pressure was <120 mmHg and diastolic <80 mmHg, which is recognized by the PTNT and ESC and ACC/AHA as the optimal blood pressure [16,17].

### 4.2. Assessment of the Influence of Socio-Demographic Factors, Family History of Obesity and Participation in Physical Activity in the Process of Body Weight Loss before and after Bariatric Surgery

The results of this study, and findings from international studies of patients undergoing obesity surgery, showed that the majority were women [25,31,37,38], who more often searched for effective obesity treatments [31,39]. However, it is more difficult for women to normalize their body weight due to the physiologically higher body fat content [40]. 

Many studies have also shown that other socio-demographic factors play an important role in patients’ decisions about surgical treatment of obesity. Memarian et al. [39] found that among respondents with lower levels of education and lower incomes, bariatric treatment was less common due to lower awareness of the risk of obesity and greater acceptance of their environment. The results of research conducted by many authors have established that a lower level of education is one of the determinants of lower effectiveness of body weight reduction in people with obesity, including non-compliance with dietary recommendations [31,41,42,43]. Literature data also indicate that lower family income is one of the socio-demographic factors associated with a higher prevalence of obesity and the resulting need for surgical treatment [44,45]. A study of the demographic structure [46] in a group of people treated surgically showed that most of the patients lived in the city, and fewer lived in the countryside. The results of our research confirm the observations of other authors. Studies involving large groups of people have shown that unemployment promoted weight gain [47], as did the younger age of bariatric patients [3]. Literature data emphasize the importance of the genetic factor in the prevalence of obesity in the family, and above all, inappropriate eating habits passed down to children by parents and shaping their nutritional and health attitudes [48,49]. In our study, obesity in parents or grandparents was declared by 83% of patients; however, respondents from families with obesity were more successful in reducing their BMI (not confirmed in the literature due to a lack of studies). Few respondents in our study declared physical activity compared to the study conducted by Soares et al. [31], in which over 60% of respondents reported regular physical activity after bariatric surgery. Based on a meta-analysis of 26 studies on physical activity after bariatric surgery, Herring et al. [5] found that during the 6 months after the procedure, physical effort is less intense, despite the fact that patients’ mobility is greater. In a group of 22 patients after bariatric surgery, Afshar et al. [50] documented that low physical activity characterized people with obesity before and after bariatric surgery, both in self-observation and in observation controlled by measuring devices. A different opinion was expressed by Coen et al. [6], who showed that physical activity most often increased after bariatric surgery along with weight loss, although the use of activity measuring devices established that patients overestimated their activity in self-observation. Coen and Goodpaster [51] emphasized that intensifying physical activity combined with reducing energy intake may be effective in maintaining weight loss after bariatric surgery, as well as improving overall health, vitality, and physical fitness [52]. There are still no precisely designed programs that would ensure the greatest benefit from exercise before and after bariatric surgery [53].

### 4.3. Assessment of Eating Habits for Weight Loss after Bariatric Surgery

Surgical treatment is generally considered the only documented treatment for class II or III obesity that leads to permanent body weight loss and resolution of obesity comorbidities and reduced risk of death [54,55,56,57]. However, after bariatric surgery, the patient should be placed under long-term dietary care because a properly balanced diet with modification of nutritional behavior (regular consumption of the recommended number of meals, control of food consumption, elimination of snacking during the day and night, use of appropriate culinary techniques with fat reduction, limitation of alcohol consumption) is required for body weight loss [2,31,58,59,60]. Hence, in this study, patients were provided with dietary care and dietary recommendations.

Regular consumption of 5–6 meals a day is associated with a lower risk of being overweight and obese [1,7,61,62], and failure to follow this rule reduces the chances of achieving proper body weight reduction and maintaining this effect for a long time. Snacking during the day increases energy intake, as frequently consumed snacks provide additional amounts of simple sugars or fats [63,64]. The research results show that the most frequently consumed snacks were high-calorie snacks with low nutritional value, which did not give the feeling of satiety and were not included in the energy balance of people with obesity [32,64,65]. Centofanti et al. [65] emphasize that eating at night impairs glucose metabolism, increases insulin resistance and inhibits fat oxidation. Furthermore, eating 4–5 meals a day at fixed times during the day ensures better metabolism and optimal use of nutrients. In addition, studies have reported that patients’ preferences changed over time after surgery. Patients with a follow-up of ≥2 years liked desserts, fried foods, fat, bread, hot drinks, and alcohol more than patients with a follow-up of <2 years, and patients who benefited more from the surgery liked green vegetables more and less starchy foods, milk, and sweet dairy products [8], which is also confirmed by the research of other authors [21].

### 4.4. Study Strengths and Limitations

The strength of this study is that the analysis took into account several factors influencing the variation of BMI in patients undergoing bariatric surgery and dietary care. To our knowledge, no study has covered as many factors, including modifiable and non-modifiable, nutritional and non-nutritional factors, in assessing BMI variation over time as our study (0–9 months after SG). Furthermore, anthropometric data and blood pressure were assessed based on the data obtained from measurements rather than self-reported data. Additionally, all measurements were conducted with a standardized procedure by a specially certified dietician, ensuring reliable results and minimizing bias.

Our study is subject to some limitations. The study was conducted with fewer men in the study group (20%), which is related to the fact that the majority of patients undergoing surgical treatment for obesity are women. The second limitation is that the study ended 9 months after the procedure due to patients opting out of further dietary consultations. Another limitation is that the tested model explained 68% of the BMI variation after SG surgery for all assessed factors. Consequently, the research requires further follow-up to describe how the consumption of selected food groups and nutrients affects body weight loss.

## 5. Conclusions

The conducted research confirmed that patients who underwent bariatric surgery, as well as dietary care after surgery, changed their lifestyle and eating habits. Patients ate meals more frequently, with shorter intervals between meals and control of food consumption, including snacking. Patients changed their cooking techniques to be more fat-free, reduced alcohol consumption, and controlled body weight more frequently. The impact of SG bariatric surgery and the time since surgery determined BMI normalization. Other factors important for BMI normalization after surgery include male gender, older age of patients, family obesity (non-modifiable factors), as well as previous forms of therapy related to weight loss before surgery, shortening the intervals between meals, and stopping eating at night (modifiable factors). The tested model explained the 68% BMI variation after SG surgery for all assessed factors.

## Figures and Tables

**Table 1 nutrients-14-05401-t001:** Characteristics of patients.

Characteristics	Total *N* = 30
Age (years)	45 ± 10 *
Sex (%*N*) women men	80 20
Place of residence (%*N*) countryside town <100.000 city >100.000	10 47 43
Education level (%*N*) primary (age 7–15 years) secondary (age 15–18 years) higher (age >18 years)	27 53 20
Professional activity (%*N*) no yes	30 70
Number of family members (%*N*) 1 2 3 4	10 33 20 37
Financial situation (%*N*) bad neither good/nor bad good	13 47 40
Family obesity (%*N*)	83
Previous forms of therapy (%*N*) no action diet diet and drug therapy	7 86 7

Data were presented as mean ± SD * and %*N* (*N* = 30) as appropriated.

**Table 2 nutrients-14-05401-t002:** Self-assessment of nutritional knowledge, providing and following nutritional recommendations, body weight control and active leisure time in patients within 9 months of therapy.

Parameters	Months after SG Surgery					*p*
	0	1	3	6	9	
	%*N* (*N* = 30)					
Self-assessment of nutritional knowledge bad neither good nor bad good	13 30 57	3 20 77	0 17 83	0 3 97	0 7 93	<0.001
Willingness to broaden nutritional knowledge	97	93	90	93	87	0.363
Obtaining nutritional recommendations, source doctor nurse dietician	80 0 10 70	100 0 0 100	100 0 0 100	100 0 0 100	100 0 0 100	<0.001 <0.001
Form of nutritional recommendations oral brochure/leaflet menus	53 70 13	100 100 100	100 100 100	100 100 100	100 100 100	<0.001 <0.001 <0.001
Content of dietary recommendations recommended and non-recommended products quantitative nutritional guidelines info on culinary techniques ready-made menus	67 60 60 20	100 100 100 100	100 100 100 100	100 100 100 100	100 100 100 100	<0.001 <0.001 <0.001 <0.001
Adherence to nutritional recommendations no adaptation to a small extent at an intermediate level to a great extent completely	33 7 27 20 13	0 3 13 37 47	7 20 30 37 7	10 13 43 30 3	20 17 37 27 0	<0.001
Number of meals 3 4–5 ≥ 6	43 54 3	7 83 10	3 90 7	10 87 3	17 73 10	<0.001
Breaks between meals 3 h 4–5 h ≥6 h	43 40 17	90 10 0	93 7 0	90 10 0	77 20 3	<0.001
Control of food consumption no sometimes yes	40 3 57	3 63 33	7 0 93	3 0 97	0 0 100	<0.001
Snacking during the day no sometimes yes	27 7 67	70 27 3	47 23 30	37 27 37	27 40 33	<0.001
Eating at night no sometimes yes	77 7 17	93 3 3	93 0 7	90 7 3	93 3 3	0.076
Culinary techniques boiling/steaming frying stewing baking	80 40 27 40	97 0 0 17	100 13 7 27	100 23 10 30	93 23 23 43	0.006 0.001 0.011 0.072
Alcohol consumption	20	0	3	3	7	0.005
Body weight control no (no/yes) 1−2×/week 3×/week 4–5×/week	27 13 40 20	13 0 50 37	6 0 57 37	10 7 63 20	10 10 57 23	0.076 <0.001
Reasons for body weight control appearance improvement of well-being concern for health	43 57 53	33 50 70	43 57 67	50 57 73	60 57 77	0.168 0.973 0.138
Active leisure time	23	20	30	27	27	0.885

Data were presented as %*N* (*N* = 30).

**Table 3 nutrients-14-05401-t003:** Anthropometric and blood pressure measurements in patients within 9 months of therapy.

Parameters	Months after SG Surgery					*p*
	0	1	3	6	9	
	*N* = 30					
Body weight (kg)	121.8 ± 19.5 119.0 88.0−170.0	108.2 ± 17.1 106.5 80.0−152.0	99.0 ± 16.8 96.0 68.0−140.0	92.5 ± 16.4 89.0 65.0−136.0	89.7 ± 16.1 87.0 62.0−135.0	<0.001
Height (cm)	167.9 ± 7.0 166.0 156.0−182.0	167.9 ± 7.0 166.0 156.0−182.0	167.9 ± 7.0 166.0 156.0−182.0	167.9 ± 7.0 166.0 156.0−182.0	167.9 ± 7.0 166.0 156.0−182.0	1.00
BMI (kg/m^2^)	43.1 ± 5.9 41.7 34.2−52.9	38.3 ± 5.1 37.8 30.7−47.0	35.0 ± 5.0 34.2 27.2−43.3	32.7 ± 4.8 32.6 25.4−42.0	31.7 ± 4.7 31.7 24.2−41.7	<0.001
Waist (cm)	127.4 ± 10.6 124.5 113.0−160.0	116.2 ± 11.7 112.0 104.0−151.0	108.1 ± 11.1 105.0 90.0−140.0	102.4 ± 12.6 100.0 85.0−136.0	100.1 ± 13.0 101.5 81.0−135.0	<0.001
Hip (cm)	138.2 ± 13.5 139.0 117.0−168.0	129.6 ± 11.3 132.5 107.0−146.0	122.1 ± 11.3 121.5 104.0−145.0	115.7 ± 10.8 116.0 97.0−141.0	112.6 ± 10.0 112.0 94.0−139.0	<0.001
WHR	0.93 ± 0.10 0.92 0.80−1.17	0.90 ± 0.09 0.89 0.76−1.13	0.89 ± 0.09 0.87 0.72−1.14	0.89 ± 0.09 0.90 0.71−1.13	0.89 ± 0.10 0.90 0.70−1.13	<0.001
Systolic pressure (mm Hg)	127 ± 14 126 110−170	119 ± 13 120 90−143	118 ± 15 120 70−140	120 ± 11 120 80−135	121 ± 12 120 80−140	0.179
Diastolic pressure (mm Hg)	83 ± 10 80 65−110	76 ± 9 80 60−100	75 ± 10 80 50−90	76 ± 8 80 50−90	78 ± 8 80 55−90	0.001

Data were presented as mean ± SD, median and min–max as appropriate.

**Table 4 nutrients-14-05401-t004:** Evaluation of anthropometric measurements and blood pressure in relation to reference values in patients within 9 months of therapy.

Parameters	Months after SG Surgery					*p*
	0	1	3	6	9	
	%*N* (*N* = 30)					
BMI (kg/m^2^) <25.0 25.0–29.99 30.0–34.99 35.0–39.99 40.0–44.99 45.0–49.99 ≥50.0	0 0 7 27 33 10 23	0 0 30 37 20 13 0	0 17 40 23 20 0 0	0 37 30 23 10 0 0	10 30 40 13 7 0 0	<0.001
WHR ≤ ref. value (F ≤ 0.8 M ≤ 1.0) > ref. value (F > 0.8 M > 1.0)	10 90	27 73	27 73	37 63	37 63	0.010
Systolic pressure (mm Hg) ≤120 >120	47 53	67 33	73 27	67 33	67 33	0.037
Diastolic pressure (mm Hg) ≤80 >80	60 40	87 13	87 13	90 10	80 20	0.001

Data were presented as %N (N = 30); ref. value F—female, M—male.

**Table 5 nutrients-14-05401-t005:** Association between BMI and factors affecting BMI variation in patients within 9 months of therapy.

Variables	Model 1	Model 2
Months after SG surgery	−1.139 *** (0.132) *−0.578*	−1.031 *** (0.124) *−0.523*
Sex		−3.957 ** (1.451) *−0.243*
Age		−0.098 * (0.048) *−0.147*
Place of residence		−0.704 (0.702) *−0.070*
Education level		0.585 (0.603) *0.061*
Professional activity		0.323 (1.026) *0.023*
Number of family members		0.122 (0.582) *0.019*
Financial situation		−0.832 (0.929) *−0.087*
Family obesity		−4.043 ** (1.244) *−0.231*
Previous forms of therapy		−7.147 *** (1.180) *−0.400*
Nutritional knowledge		−0.767 (0.910) *−0.057*
Adherence to nutritional recommendations		−0.064 (0.346) *−0.012*
Number of meals		−0.722 (0.622) *−0.077*
Breaks between meals		1.329 * (0.671) *0.159*
Control of food consumption		0.523 (0.721) *0.053*
Snacking during the day		−0.370 (0.425) *−0.049*
Eating at night		1.527 * (0.685) *0.123*
Alcohol consumption		0.186 (1.615) *0.007*
The frequency of body weight control		−0.654 (0.452) *−0.093*
Active leisure time		−0.938 (0.797) *−0.063*
Constant	40.510 (0.666)	69.818 (6.292)
R^2^	0.334	0.676

Main entries are unstandardized coefficients; numbers in parentheses are standard errors; numbers in italics are standardized coefficients (beta coefficients); * *p* < 0.05; ** *p* < 0.01; *** *p* < 0.001.

## Data Availability

The dataset used and/or analyzed during this study is available from the corresponding author upon reasonable request.

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
