# Peer review of "Environmental Factors Determining Body Mass Index (BMI) within 9 Months of Therapy Post Bariatric Surgery—Sleeve Gastrectomy (SG)"

_nutrients, 2022, doi:10.3390/nu14245401_

Round 1
Reviewer 1 Report
The authors present the important topic of bariatric surgery intervention and control in a scientific world where nutritional intervention are more often conducted through the very simple and easy practice then the overall lifestyle changes.
Reviewer 2 Report
Title: No acronyms should be included. Also, the title is very long. Recommend shortening to the main point. Also includes BP in the title but only diastolic was significant and then not included in the full models?
Abstract: Also shouldn’t include acronyms. Should include a background section/sentence.
9 months after surgery isn’t a long time. Most changes are seen within a year window after surgery.
Line 16: family “history” of obesity.
Not sure if all measures need to be included in the abstract with such detail. Only those main highlights would be helpful to see.
No conclusion statement made. Would decrease the wording for all methods and results and tell main points/highlights.
Introduction:
Would watch wording of several statements throughout the manuscript. Stating that “obesity requires treatment, including surgery” isn’t necessarily true.
Description of MBS is warranted.
There are several studies that examine obesity, in general, and related environmental, social, cultural, behavioral, etc factors.
Methods:
More information about nutrition information provided by dietitians is needed. The same information at each post-op visit? Different? Was it equal between all participants?
No family history of obesity is aunts, uncles, cousins?
Line 100: cafeteria?
Were self-reported questions validated measures? Or that the team came up with themselves?
Line 121: seems that it should be height, not growth? They aren’t currently still growing
Table 1 seems to also have percentages in it. Need to indicate which are which type of data inside the table.
Line 173: doesn’t seem like there is qualitative data in this manuscript? It is mentioned in here though.
Results:
Numbers in the tables are unclear; it seems like just providing the value or “%” would be easy to do. Also the formatting of the header and all the sub-values being centered in each cell is hard to read/interpret because it runs together – would divide the header and sub values or left justify the header portion (i.e. Financial Situation to left, values in center).
Also some of the values seem arbitrary – what does “bad” financial situation mean; what is professional activity?
Results throughout seem typical for the first months following surgery, not necessarily novel? How is this adding different or important information to the collective?
Acronym of WHR is never written out – assuming Waist to Hip Ratio?
Typically weight after surgery is usually reflected as Percent Excess Weight
Loss (%EWL) because actual body weight is such a range by race, ethnicity, age, sex, etc – not really fair to show it as kgs but rather should show as %EWL at a “healthy” BMI. Able to then compare to other bariatric surgery outcome literature.
Should age and sex be controlled for or actual variables in the model?
Seems like there are many variables included in the linear regression models that were not tested against weight in initial exploratory analysis – why were they then put into this Model 2?
Still overall wondering the novelty of this study; we know behavioral factors influence weight and lifestyle; after MBS, weight of course decreases on average (and significantly within the first year)
Discussion:
Line 227 – would caution the use to “abstaining from eating at night” – this could get into eating disorder territory if shared as a “healthful behavior” in future clinical practice. In the realm of nutrition and dietetics that can be a risky behavior. If you’re eating your caloric range in the hours your awake, it should be fine – abstaining from eating just because of a certain hour, can be dangerous. I wouldn’t use this language.
Again, I think it’s hard to see how this is contributing to the literature; the discussion shows all the studies that have looked at these variables before.
Limitations of “small group of men” need to be clarified – this makes it seem like the cohort in full, were men. Would rephrase to indicate that of the cohort, 20% were men.
The final concluding sentence in the discussion leaves readers to think that the work is about nutrition and food groups only, not other factors.
